# Anti-Hair Loss Effect of Veratric Acid on Dermal Papilla Cells

**DOI:** 10.3390/ijms26052240

**Published:** 2025-03-02

**Authors:** Jiyoung You, Youngsu Jang, Junbo Sim, Dehun Ryu, Eunae Cho, Deokhoon Park, Eunsun Jung

**Affiliations:** Biospectrum Life Science Institute, Sinsu-ro, Suji-gu, Yongin-City 16827, Gyeonggi-Do, Republic of Korea; biotr@biospectrum.com (J.Y.); biogc@biospectrum.com (Y.J.); bioba@biospectrum.com (J.S.); biosc@biospectrum.com (D.R.); biozr@biospectrum.com (E.C.); pdh@biospectrum.com (D.P.)

**Keywords:** veratric acid, hair growth, dermal papilla cells, hair inductivity, senescence

## Abstract

The activation of hair follicle dermal papilla cells (HFDPCs), a critical target of hair loss relief, can be achieved through the upregulation of proliferation, the stimulation of hair inducibility, and the inhibition of cellular senescence. Veratric acid (VA) is a major benzoic acid found in fruits and vegetables. The biological activity of VA on HFDPCs remains to be elucidated. In this study, we investigated the capacity of VA for hair loss mitigation. An MTT assay, Ki67 staining, quantitative RT-PCR (qRT-PCR), and a Western blot analysis were performed to confirm the proliferative effect of VA. Hair inductivity was determined through a cell aggregation assay and ALP staining. Annexin V/PI staining was performed to confirm the anti-apoptotic effect of VA. The inhibitory effect of VA on cellular senescence was confirmed by a β-galactosidase (β-gal) assay and qRT-PCR using replicative senescence and oxidative stress-induced senescence models. As a result, VA dose-dependently upregulated the proliferation of HFDPCs, the expression of growth factors, and β-catenin protein levels. VA also dose-dependently increased ALP activity and cell aggregation and decreased apoptotic cells through the regulation of *BCL2* and *BAX* expression. Moreover, VA reduced β-gal activity and the senescence-associated secretory phenotype (SASP) in a dose-dependent manner in senescent HFDPCs. These findings suggest that VA may serve as a potential therapeutic agent for alleviating hair loss by targeting multiple pathways involved in HFDPC activation.

## 1. Introduction

Although hair loss is not a fatal disease to life, it causes a negative social impression and psychological pain. Factors related to hair loss include heredity, stress, hormones, drugs, nutritional imbalance, infection, and chemical irritation. As these factors often interact, identifying the underlying causes and developing effective treatments for hair loss remains challenging [1]. Currently, strategies aimed at stimulating hair follicle-associated cells are being actively investigated as potential therapeutic approaches for hair loss [2].

Hair is produced from hair follicles anatomically linked to sebaceous glands and arrector pili muscles [3]. The interaction of cells that make up hair follicles can regulate the hair cycle through the anagen, catagen, and telogen phases, leading to hair growth and degeneration. Hair follicle dermal papilla cells (HFDPCs) are key regulatory cells within hair follicles (HFs) that modulate the hair cycle through interactions with stem cells residing in the bulge region [4]. Moreover, the number of HFDPCs is positively correlated with hair shaft thickness. A decrease in the number of DPCs can lead to failure of anagen re-entry of HFs, resulting in hair loss [5]. Hair inductivity also plays a key role in the development and regeneration of hair follicles. Hair-inducing properties can be restored in three-dimensional (3D) spheroid cultures, starting with self-aggregation of hair follicles mediated by TGF-β2 [6,7]. DPC undergoes apoptosis in the catagen phase, leading to hair loss. Thus, the inhibition of DPC apoptosis can contribute to hair loss relief [8].

HFDPCs in the bald regions of patients with androgenetic alopecia (AGA) exhibit characteristics of premature senescence and demonstrate greater sensitivity to environmental stressors compared to HFDPCs from non-balding areas [9]. HFDPC senescence not only impairs the proliferative capacity of dermal papilla cells (DPCs) but also disrupts hair follicle formation, contributing to hair follicle miniaturization and subsequent hair loss. Replicative senescent DPCs show decreases in cell viability and the S phase in the cell cycle [10]. In an oxidative stress-induced premature senescence model, the conversion from the telogen to anagen phases is suppressed due to DPC senescence [11].

Veratric acid (VA, 3,4-dimethoxybenzoic acid, CAS no. 93-07-2) is a major simple benzoic acid found in fruits, vegetables and mushrooms [12]. VA has been reported to possess anti-inflammatory and antioxidant activities. VA inhibits the production of nitric oxide (NO), interleukin-6 (IL-6), and interferon-gamma (IFN-γ) in LPS-stimulated RAW264.7 cells and protects keratinocytes from oxidative stress by restoring UVB-induced depletion of glutathione (GSH) [13,14]. Inflammatory cytokines, such as TNF-α and IL-1β, activate the NF-κB signaling pathway, which can suppress Wnt/β-catenin signaling, a key regulator of hair growth, and induce apoptosis in hair follicle stem cells [15,16]. Additionally, oxidative stress inhibits Wnt/β-catenin signaling and promotes senescence in DPCs by activating p16 and p53, thereby disrupting the hair growth cycle [11,17]. Therefore, the previously reported anti-inflammatory and antioxidant effects of VA are expected to suppress inflammatory mediators and reduce oxidative stress, thereby protecting Wnt/β-catenin signaling, inhibiting apoptosis and senescence, and ultimately promoting DPC activation and hair growth.

Furthermore, VA has been reported to exert anti-inflammatory and anti-apoptotic effects, which may contribute to hair follicle (HF) maintenance and hair loss prevention. Previous studies suggest that VA reduces inflammation by suppressing CXCL1, a pro-inflammatory chemokine secreted from dermal white adipose tissue (dWAT), which is known to delay HF cycling, induce premature catagen transition, and promote HF miniaturization [18]. Additionally, CXCL1 has been shown to induce apoptosis in the hair matrix and connective tissue sheath, further impairing follicle maintenance. By inhibiting CXCL1 expression, VA may help preserve HF integrity and support hair growth. However, it remains unclear whether VA directly influences HFDPCs to promote hair growth. Therefore, this study aimed to investigate the effects of VA on HFDPCs. We found that VA enhanced the proliferation and hair-inductive potential of HFDPCs while attenuating cellular senescence.

## 2. Results

### 2.1. Veratric Acid Increases Proliferation of HFDPCs

HFDPCs function as a niche for progenitor cells during the growth and resting phases of the hair cycle. Thus, HFDPC proliferation influences both the formation and regeneration of hair follicles [7]. To assess the effect of VA on the proliferation of HFDPCs, cells were cultured with 10, 25, or 50 μM VA. The proliferative effects of VA were evaluated using MTT and Ki67 staining assays. As shown in Figure 1A, treatment with 50 μM VA increased cell proliferation by up to 18% compared to the control group. This increase was comparable to the 16% increase observed with minoxidil, which was used as a positive control. The proportion of Ki67-positive cells increased by 24% using 50 μM VA compared to the control group (Figure 1B). We then investigated the effect of VA on the expression of genes involved in the regulation of DPC proliferation. As shown in Figure 1C, VA treatment upregulated the expression of *LEF1* and *CCND1* in DPCs. LEF1, a key transcription factor in the Wnt/β-catenin pathway, directly regulates *CCND1* expression, which promotes cell cycle progression by activating CDK4/6 [19]. This activation leads to Rb phosphorylation, releasing E2F, which drives the expression of S-phase genes, facilitating HFDPC proliferation. In addition, VA increased the expression of β-catenin and PCNA (Figure 1D), both of which are associated with cell proliferation. These findings suggest that VA promotes HFDPC proliferation through the Wnt/β-catenin signaling pathway by enhancing the expression of *LEF1* and *CCND1*. However, it remains unclear whether VA directly activates Wnt signaling or indirectly stabilizes β-catenin, requiring further investigation.

### 2.2. Veratric Acid Promotes Expression of Growth Factors Involved in Hair Growth

Representative growth factors for hair growth include VEGF, EGF, IGF1, and HGF. To determine whether VA enhances the expression of these growth factors, we observed their expression levels. As a result, VA upregulated major growth factors related to hair growth, such as *VEGFA*, *EGF*, *IGF1*, and *HGF*, in HFDPCs in a concentration-dependent manner. Specifically, the *VEGFA*, *EGF*, *IGF1*, and *HGF* expression levels increased by 3-, 2.5-, 4-, and 5-fold using 50 μM VA, respectively (Figure 2). Given that Wnt/β-catenin signaling plays a crucial role in regulating DPC function and growth factor expression, VA-induced β-catenin activation may contribute to the observed increase in these factors [20,21]. These findings suggest that VA enhances the expression of hair growth factors by activating the Wnt/β-catenin signaling pathway, thereby promoting hair follicle growth.

### 2.3. Veratric Acid Improves Hair Inductivity

Self-aggregation is a prerequisite for HFDPCs to induce human hair follicle formation and is closely associated with hair follicle development and hair growth induction [6]. HFDPCs possess an intrinsic ability to self-aggregate and contribute to hair structure formation. Moreover, the expression of *TGFB2* has been identified as a key regulator of HFDPC self-aggregation [7]. To evaluate the relationship between VA treatment and hair inductivity, we measured ALP activity and conducted 3D aggregation assays. As shown in Figure 3A, the ALP staining levels significantly increased in the VA-treated groups compared to the control group. The ALP levels increased by approximately 60% using 50 μM VA. ALP is known to be associated with hair induction and reconstruction of the bulbar structure of hair [22]. Additionally, VA enhanced HFDPC self-aggregation in a concentration-dependent manner, as confirmed by the 3D culture aggregation assay (Figure 3B). At 50 μM, VA exhibited an aggregation capacity comparable to that of TGF-β2, which was used as a positive control. Furthermore, VA upregulated *TGFB2* expression in a concentration-dependent manner (Figure 3C). Given that TGF-β2 plays a key role in promoting DPC self-aggregation, the observed increase in *TGFB2* expression following VA treatment suggests that VA facilitates self-aggregation through *TGFB2* upregulation. These findings suggest that VA contributes to the restoration of hair-inductive activity by increasing ALP activity and promoting HFDPC aggregation through the upregulation of *TGFB2*.

### 2.4. Veratric Acid Reduces Apoptosis of HFDPCs

During the catagen phase, apoptosis of HFDPCs is accelerated, contributing to hair loss [23]. Minoxidil has been reported to exert anti-hair loss effects by inhibiting HFDPC apoptosis [8,24]. To determine whether VA reduces HFDPC apoptosis, we examined the expression levels of *BCL2*, an anti-apoptotic factor, and *BAX*, a pro-apoptotic factor. As shown in Figure 4A, VA upregulated *BCL2* expression while downregulating *BAX* expression. BCL2 is an anti-apoptotic protein that inhibits mitochondrial outer membrane permeabilization (MOMP), preventing cytochrome c release and caspase activation, thereby blocking apoptosis. In contrast, BAX is a pro-apoptotic protein that promotes MOMP, leading to cytochrome c release and caspase-mediated apoptosis. The modulation of these key apoptotic regulators suggests that VA inhibits apoptosis by shifting the balance towards cell survival [25]. In addition, the proportion of apoptotic cells was assessed using a FACS analysis with PI and Annexin V staining. The results demonstrated that treatment with 50 μM VA reduced the proportion of apoptotic cells by 30% compared to the control group, which was set to 100% (Figure 4B). This result indicates that VA-mediated regulation of *BCL2* and *BAX* contributes to the suppression of apoptosis in HFDPCs.

### 2.5. Veratric Acid Inhibits Senescence in Replicative Senescent HFDPCs

A representative characteristic of replicative senescent cells is the arrest of cell proliferation with an increase in SA-β-Gal activity and the senescence-associated secretory phenotype (SASP) such as inflammatory cytokines [26]. IL-6 is a major proinflammatory cytokine of SASP, a mediator of senescent cells [27,28]. A previous study has reported that the expression level of IL-6 is higher in balding DPC than in its non-balding DPC, and IL-6 can inhibit hair shaft elongation in hair organ culture [29]. In a model of replicative senescent dermal cells, TGF-β1 treatment increased the production of SASPs, including IL-6 and IL-8. TGF-β1 secreted from DPCs plays a major role in impairing hair growth by regulating the hair cycle and inducing hair follicle miniaturization, which is associated with the progression of androgenetic alopecia (AGA) [30,31]. To confirm the inhibitory effect of VA on the cellular senescence of HFDPCs, we assessed β-gal activity and the expression of senescence-associated factors, including *IL6* and *TGFB1*. As shown in Figure 5A,B, VA reduced β-gal activity and suppressed the expression of *IL6* and *TGFB1*. Treatment with 50 μM VA decreased β-gal activity by 55% compared to the control group of senescent HFDPCs (Figure 5A). The expression levels of *IL6* and *TGFB1* were reduced by 38% and 37%, respectively, following treatment with 50 μM VA (Figure 5B). These findings suggest that VA attenuates cellular senescence by suppressing senescence-associated factors in the replicative senescent HFDPC model.

### 2.6. Veratric Acid Alleviates Oxidative Stress-Induced Senescence of HFDPCs

Oxidative stress-induced premature senescence of HFDPCs disrupt epithelial–mesenchymal interactions and lead to hair follicular senescence. H_2_O_2_-induced DPC senescence fails to respond to keratinocytes stimulation, thereby inhibiting hair follicle stem cell growth and ultimately leading to a loss of inductive capacity for follicle neogenesis [11]. p21 is a key cell cycle inhibitor that promotes senescence by halting G1/S and G2/M transitions, and its expression is upregulated by oxidative stress through p53 activation [32]. Several studies have reported that H_2_O_2_ treatment increases p21 and β-gal activity in DPCs [17,33]. Additionally, TGF-β1 is induced by androgen in DPCs and this process is ROS-mediated [30]. In an H_2_O_2_-induced senescent HFDPC model, treatment with 50 μM VA inhibited β-gal activity to 49% compared to the H_2_O_2_-treated control group, which was set to 100% (Figure 6A). Moreover, VA downregulated the expression levels of *P21* and *TGFB1*, which were upregulated by H_2_O_2_ (Figure 6B). These results suggest that VA exerts an inhibitory effect on oxidative stress-induced DPC senescence by suppressing key senescence-associated factors, including *P21* and *TGFB1*.

## 3. Discussion

The number of HFDPCs is positively correlated with hair thickness and a decrease in HFDPCs is associated with the delayed initiation of anagen and retention in the telogen phase. Maintaining the proliferative capacity of HFDPCs is a critical target for hair loss treatment [34,35]. In this study, VA promoted cell proliferation and increased β-catenin protein levels in HFDPCs. The Wnt/β-catenin signaling pathway activates the TCF/LEF transcription factor, inducing *CCND1* gene expression, and maintains the expression of DP marker genes, including *ALP*, *LEF1*, and *VCAN*, in DPCs [36,37]. PCNA is a proliferation marker that is expressed in the DNA synthesis phase. Our results show that VA upregulated *LEF1* and *CCND1* expression and increased PCNA protein levels in HFDPCs.

In this study, we demonstrated that VA promotes HFDPC proliferation via Wnt/β-catenin signaling, as evidenced by the upregulation of *LEF1*, *CCND1*, β-catenin, and PCNA. Given the critical role of Wnt/β-catenin signaling in hair follicle regeneration, these findings suggest that VA may support hair growth by reinforcing the proliferative capacity of DPCs. Additionally, Wnt/β-catenin signaling is known to regulate not only cell proliferation but also dermal–epidermal interactions essential for hair follicle development [38]. By enhancing Wnt/β-catenin signaling, VA may contribute to hair follicle maintenance and anagen induction. Moreover, Wnt/β-catenin signaling plays a key role in hair cycle transitions by sustaining the anagen phase and promoting telogen-to-anagen transition [21]. The VA-induced activation of Wnt/β-catenin signaling may thus contribute to maintaining the hair growth phase and facilitating its initiation. However, the precise molecular mechanism by which VA influences this pathway remains unclear, and further studies are needed to determine whether VA affects upstream regulators of Wnt/β-catenin signaling or facilitates β-catenin nuclear translocation to activate target gene expression.

Various growth factors regulate the hair cycle, among which VEGF, EGF, IGF1, and HGF are key factors known to promote the transition from telogen to anagen. *VEGF* is highly expressed in HFDPCs and serves as a major regulator of vascular growth, inducing HFDPC proliferation [39]. EGF has been shown to enhance HFDPC proliferation and regulate the expression of *ALP* or *IGF* in HFDPCs [40]. Additionally, EGF promotes ORS proliferation and migration by stimulating β-catenin translocation and upregulating *SOX9* expression [41]. IGF-1 promotes hair follicle growth by sustaining the anagen phase. HGF secreted by HFDPCs acts on follicular epithelial cells to stimulate hair growth [42,43]. In this study, VA upregulated the expression of *VEGFA*, *EGF*, *IGF1*, and *HGF* in HFDPCs, suggesting its potential role in promoting hair follicle growth. Given that Wnt/β-catenin signaling plays a crucial role in regulating growth factor expression in dermal papilla cells, VA-induced β-catenin activation may contribute to the observed increase in these growth factors. Previous studies have demonstrated that Wnt/β-catenin signaling enhances *VEGF*, *EGF*, and *IGF1* expression, which are essential for hair follicle development and regeneration [21]. Thus, VA may promote the upregulation of growth factors by activating the Wnt/β-catenin pathway, thereby facilitating hair follicle growth and maintenance. These findings suggest that VA contributes to hair follicle regeneration not only by directly enhancing HFDPC proliferation but also by upregulating key growth factors through Wnt/β-catenin signaling.

Follicle regeneration represents a promising target for cell therapy in hair loss treatment. The HF-inducing ability of DPCs is enhanced under aggregate conditions. In mice, self-aggregated DPCs exhibit characteristics similar to those of embryonic cells involved in follicle neogenesis [44,45,46]. Previous studies have demonstrated that *ALP* overexpression enhances follicle induction by regulating the Wnt/β-catenin signaling pathway and genes associated with hair inductivity. Moreover, ALP activity is restored in the spheroid cultures of DPCs, supporting the importance of self-aggregation in hair induction [47]. Three-dimensional spheroid culture conditions have been shown to enhance the hair-inducing properties of DPCs [48]. In this study, we found that VA treatment increased ALP activity and enhanced self-aggregation in spheroid-cultured DPCs. Notably, VA also upregulated *TGFB2* expression in a concentration-dependent manner. Given that TGF-β2 is a critical regulator of HFDPC self-aggregation, the observed enhancement of aggregation by VA treatment suggests that VA promotes follicle regeneration by facilitating DPC self-aggregation via *TGFB2* upregulation. These findings indicate that VA may enhance the hair-inductive properties of HFDPCs through multiple mechanisms, including ALP activation and *TGFB2*-mediated self-aggregation, ultimately supporting follicle regeneration.

During the hair cycle, cell proliferation and apoptosis occur in a cyclic manner, with apoptosis being predominantly observed during the catagen phase [8]. However, DPCs are considered resistant to apoptosis due to the continuous expression of the anti-apoptotic gene *BCL2* throughout the hair cycle [49]. Our findings demonstrated that VA treatment increased *BCL2* expression while decreasing *BAX* expression in HFDPCs, a pattern similar to that observed with minoxidil treatment. These results suggest that VA may enhance cell survival and attenuate apoptosis in DPCs by modulating the *BCL2/BAX* ratio.

Cellular senescence is characterized by irreversible cell cycle arrest, which is associated with an increase in the number of non-dividing cells and prolonged cell division time as proliferative capacity declines [50]. A previous study reported that replicative senescence in HFDPCs is characterized by reduced cell viability, decreased S-phase proportion in the cell cycle, and impaired migratory capacity, along with increased β-gal activity compared to young DPCs (passage 3) [10]. IL-6 and TGF-β1 are key components of the SASP, contributing to cellular senescence and inflammation. Our results showed that VA significantly suppressed *IL6* and *TGFB1* expression in senescent HFDPCs, suggesting its potential role in regulating senescence-related pathways. *IL6* expression is primarily controlled by NF-κB, which remains persistently active in senescent cells [51]. Chronic NF-κB activation drives IL-6 secretion, exacerbating senescence. Thus, VA may mitigate senescence by suppressing NF-κB signaling. TGF-β1 is a key regulator of SASP and senescence, activating the Smad2/3 pathway to induce cell cycle arrest [52]. Smad2/3 activation promotes *P16* and *P21* expression, reinforcing senescence. The VA-mediated suppression of *TGFB1* suggests that VA may disrupt Smad signaling, preventing senescence progression. Oxidative stress is also a major contributor to HFDPC senescence. It has been reported that in DPCs undergoing premature senescence induced by H_2_O_2_, the ability to aggregate is diminished, and the proliferation of co-cultured keratinocytes is also suppressed [11]. Moreover, DPCs derived from androgenetic alopecia (AGA) patients exhibit higher β-gal activity and reduced cell motility compared to non-balding DPCs under oxidative stress, suggesting that oxidative stress may play a role in the pathogenesis of AGA [53]. In this study, VA treatment significantly reduced oxidative stress-induced senescence markers, suggesting its protective role in maintaining DPC function. p21, a key mediator of p53-dependent cell cycle arrest, promotes senescence by halting cell cycle progression at the G1/S and G2/M transitions [32]. VA downregulated *P21* expression, indicating its potential to prevent oxidative stress-induced growth arrest. Additionally, TGF-β1 reinforces SASP and sustains the senescent phenotype by amplifying senescence-associated signaling [30]. The suppression of *TGFB1* by VA suggests that it may alleviate senescence by disrupting this feedback loop, thereby preventing the reinforcement of senescence-associated pathways. Thus, VA appears to exert dual protective effects by targeting distinct regulatory pathways in different senescence models. Taken together, these findings suggest that VA protects HFDPCs from both replicative and oxidative stress-induced premature senescence through distinct mechanisms. While VA appears to suppress SASP-driven pathways (NF-κB and TGF-β1) in replicative senescence, it modulates p53-p21 and TGF-β1 signaling to counteract oxidative stress-induced senescence. By preserving the proliferative and hair-inductive potential of HFDPCs, VA may contribute to maintaining hair follicle homeostasis and delaying follicular aging.

In this study, HFDPCs were used as the primary in vitro model to evaluate the efficacy of VA against hair loss. HFDPCs play a crucial role in hair growth regulation and dermal–epidermal interactions, making them a widely recognized and reliable model in hair follicle biology research. However, to further assess its practical applicability, additional studies using 3D spheroid cultures, ex vivo follicle organ cultures, and in vivo models will be necessary.

## 4. Materials and Methods

### 4.1. Cell Culture

The human follicular dermal papilla cells (HFDPCs) were purchased from Promocell (Heidelberg, Germany) and cultured at 37 °C in a 5% CO_2_ incubator using a follicle dermal papilla cell growth medium mixed with a growth medium supplement mix (Promocell, Heidelberg, Germany). HFDPC maintenance was performed using a follicle dermal papilla cell growth medium, while all experiments involving veratric acid (VA, Sigma-Aldrich, St. Louis, MO, USA) treatment were conducted in Dulbecco’s modified eagle medium (DMEM, Hyclone, Logan, UT, USA) supplemented with 1% fetal bovine serum (FBS, Gibco, Carlsbad, CA, USA) and penicillin/streptomycin (Gibco, Carlsbad, CA, USA). All experiments, except for the replicative senescence model, were conducted using HFDPCs at passages 3 or 4.

### 4.2. Ethical Statement

The human follicular dermal papilla cells (HFDPCs) used in this study were purchased from PromoCell (Heidelberg, Germany). PromoCell ensures that all human primary cells are ethically sourced in compliance with international ethical guidelines, including the Human Tissue Act (HT Act) of 2004. All cells were obtained with informed consent from donors and processed according to regulatory standards. Therefore, separate ethical approval was not required for this study.

### 4.3. Cell Viability Assay

Cell viability was determined with a 3-(4,5-dimethyl-2-thiazoyl)-2,5-diphenyl-2H-tetrazolium bromide (MTT, Sigma-Aldrich, St. Louis, MO, USA) assay. The HFDPCs were treated with VA or minoxidil (Sigma-Aldrich, St. Louis, MO, USA, a positive control) for 72 h. Minoxidil has been widely used as a positive control for DPC proliferation assays in previous studies [24,54]. VA or minoxidil was solubilized in DMEM (Hyclone, Logan, UT, USA) supplemented with 1% FBS (Gibco, Carlsbad, CA, USA) and 1% penicillin/streptomycin (Gibco, Carlsbad, CA, USA) at 37 °C in a 5% CO_2_ incubator. The cultured cells were treated with 100 μg/mL MTT for 2 h. Formazan was dissolved in dimethyl sulfoxide (DMSO, Sigma-Aldrich, St. Louis, MO, USA) to measure cell viability at 570 nm using a spectrophotometer (Epoch, Bio-Tek Inc., Winooski, VT, USA). The mean absorbance of the control (NT) group was set to 100%, and all measured absorbance values were normalized to this standard and expressed as a percentage.

### 4.4. Cell Proliferation Assay

Proliferation was measured with Ki67 (Ki-67 Monoclonal Antibody, FITC, eBioscience^TM^, Thermo Fisher Scientific, Waltham, MA, USA) staining. The HFDPCs were cultured with VA for 24 h and washed using 1X phosphate-buffered saline (PBS, Welgene, Gyeongsan, Republic of Korea). After the collected cells were fixed with ice-cold 70% EtOH for 16 h at −20 °C, EtOH was removed by spinning down the cells. The cells were stained with FITC-conjugated anti-Ki67 antibody at room temperature (RT) for 1 h. Fluorescent level was analyzed using a BD Accuri^®^ C6 flow cytometer (BD Biosciences, San Jose, CA, USA) with a blue laser at 488 nm.

### 4.5. Quantitative Real-Time PCR

Total RNA was purified using a Trizol reagent (Thermo Fisher Scientific, Waltham, MA, USA). Briefly, the cultured cells were treated with Trizol and harvested. The upper part separated by chloroform (Sigma-Aldrich, St. Louis, MO, USA) was mixed with 100% ice-cold isopropyl alcohol (DAEJUNG, Siheung, Republic of Korea). The mixture was then centrifuged to remove the supernatant. The RNA pellet was washed with 70% EtOH in DEPC-treated water. Total RNA was measured using the Gen5 microplate reader software ver. 2.09 (Bio-Tek Inc., Winooski, VT, USA) with the nucleic acid quantification function. The RNA purity was determined by the A260/A280 ratio, with an average value of 1.98. Purified total RNA (1 μg) was subjected to cDNA synthesis using an amfiRivert cDNA Synthesis Platinum Master Mix (GenDEPOT, Barker, TX, USA) with a TaKaRa Thermal Cycler Dice Touch (TP350, TAKARA BIO INC., Otsu, Shiga, Japan). Gene expression was measured with a 7500 Real-Time PCR System (Applied Biosystems, Thermo Fisher Scientific, Waltham, MA, USA) using an AMPIGENE^®^ qPCR Green Mix Hi-ROX (Enzo Life Sciences, Farmingdale, NY, USA) and primers ordered from Bioneer Corp. (Daejeon, Republic of Korea). The sequences of primers used are shown in Table 1. The qRT-PCR cycling conditions consisted of initial denaturation at 95 °C for 10 min, followed by 40 cycles of denaturation at 95 °C for 15 s and annealing/extension at 60 °C for 35 s. After amplification, a dissociation curve analysis was conducted with the following conditions: 95 °C for 15 s, 60 °C for 1 min, and 95 °C for 15 s. Fluorescence signals were recorded at the end of each cycle, and melt curve data were collected during the dissociation phase to confirm the specificity of the amplified products.

### 4.6. Western Blot Analysis

For the Western blot analysis of β-catenin and PCNA, the HFDPCs were seeded into a 100 mm dish and cultured with VA for 24 h or 48 h. The cells were lysed with lysis buffer (PRO-PREP, iNtRON Biotechnology, Seongnam, Republic of Korea) and centrifuged at 13,000 x g for 10 min. The proteins were quantified by a Bradford assay and separated by sodium dodecyl sulfate-polyacrylamide gel electrophoresis (SDS-PAGE). The separated proteins were transferred to nitrocellulose membranes with a transfer kit (iBlot2, iBlot gel transfer stacks, Thermo Fisher Scientific, Waltham, MA, USA) and blocked with 5% skin milk at RT for 1 h. The membrane was incubated with a β-catenin antibody (1:1000), a PCNA antibody (1:1000) (Cell Signaling Technology, Danvers, MA, USA), and a GAPDH antibody (1:1000) (Santa Cruz Biotechnology, Inc., Dallas, TX, USA) at 4 °C overnight. After washing with PBST, the membrane was incubated with a mouse or rabbit IgG secondary antibody (1:1000, Cell Signaling Technology, Danvers, MA, USA) at RT for 1 h. Proteins were developed with a chemiluminescence substrate (GloBrite ECL Reagent Kit, R&D System, Detroit, MI, USA) and normalized to GAPDH.

### 4.7. Aggregation Assay

An aggregation assay was conducted using ultra-low attachment multiple well plates (ULA, Corning^®^ Costar^®^, Corning, NY, USA). The HFDPCs were seeded into ULA plates at a density of 2 × 10^4^ cells per well with VA and cultured for ~72 h. Cell aggregation was observed under a microscope (EVOS^®^ FL, Microscope, Thermo Fisher Scientific, Waltham, MA, USA).

### 4.8. Alkaline Phosphatase Staining

The alkaline phosphatase (AP) level was detected using a BCIP/NBT Color Development Substrate (5-bromo-4-chloro-3-indolyl-phosphate/nitro blue tetrazolium, Promega, WI, USA). The HFDPCs cultured with VA for 72 h were treated with an alkaline phosphatase buffer (100mM Tris-HCl [pH 9.0], 150 mM NaCl, 1 mM MgCl_2_) added to a substrate solution containing NBT and BCIP for 2 h. After images of the stained cells were taken with a microscope (Nikon ELWD 0.3/OD75 Microscope Condenser, Tokyo, Japan, Tucsen TCH-5.0 CCD camera and ISCapture program, Tusen, Fuzhou, China), the AP levels were measured using ImageJ software version 1.52a (National Institutes of Health, Bethesda, MD, USA).

### 4.9. Apoptosis

Programmed cell death was measured using annexin V and propidium iodide (PI). The cultured cells were harvested and washed using ice-cold 1XPBS. After the cells were stained with Alexa Fluor^TM^ 488 Annexin V and PI working solution included in an Apoptosis kit (Dead Cell Apoptosis Kit with Annexin V Alexa Fluor^TM^ 488 & Propidium Iodide, Invitrogen, Carlsbad, CA, USA), they were incubated at RT for 10 min. The stained cells were analyzed with a BD Accuri^®^ C6 flow cytometer (BD Biosciences, San Jose, CA, USA) at 488 nm.

### 4.10. Replicative and H_2_O_2_-Induced Senescence

The HFDPCs used in the replicative senescence model were at passages 9 to 12. They were obtained through repeated subculture of the cells. The HFDPCs used in the hydrogen peroxide (H_2_O_2_)-induced senescence model were at passage 3 or 4. Senescence was induced by pretreating the cells with 300 μM H_2_O_2_ (Sigma-Aldrich, St. Louis, MO, USA) for 2 h.

### 4.11. SA β-Galactosidase Staining

Senescence level was determined with a β-galactosidase assay. Cultured replicative or H_2_O_2_-induced senescence HFDPCs incubated with VA were washed with PBS and harvested. After the cells were stained with SPiDER-β-gal (Dojindo Laboratories, Kumamoto, Japan) at RT for 30 min, the β-gal level was measured with a BD Accuri^®^ C6 flow cytometer (BD Biosciences, San Jose, CA, USA).

### 4.12. Statistical Analysis

All experiments were repeated at least three times. The data were analyzed using Student’s *t*-test. The results obtained are expressed as mean ± standard deviation (S.D.), and a *p*-value of less than 0.05 was considered statistically significant.

## 5. Conclusions

This study is the first to demonstrate that VA directly activates human hair follicle dermal papilla cells (HFDPCs) by regulating multiple pathways involved in hair growth. The results showed that VA promoted the proliferation of HFDPCs and upregulated growth factors associated with hair growth. Furthermore, VA enhanced ALP activity, facilitated cell aggregation, and increased the expression of key factors involved in follicular inductivity, while concurrently suppressing apoptosis in HFDPCs. In addition, VA attenuated both replicative senescence and oxidative stress-induced senescence in HFDPCs by downregulating senescence-associated factors. Collectively, these findings suggest that VA has the potential to mitigate hair loss by promoting HFDPC proliferation and suppressing cellular senescence (Figure 7).

## Figures and Tables

**Figure 1 ijms-26-02240-f001:**
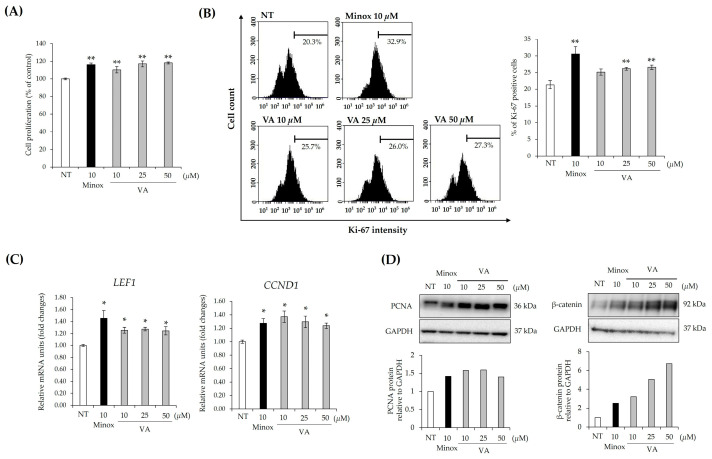
Veratric acid (VA) induces proliferation and growth signaling factors in HFDPCs. (**A**) Cell viability measured by MTT after HFDPCs were cultured with VA or minoxidil for 72 h. (**B**) Under the same condition for 48 h, cell proliferation was confirmed using Ki67 staining. (**C**) *CCND1* and *LEF1* mRNAs in HFDPCs treated with VA or minoxidil. (**D**) Western blot images of PCNA and β-catenin. All values were measured in triplicate (n = 3). Data are presented as mean ± SD. *, *p* < 0.05; **, *p* < 0.01.

**Figure 2 ijms-26-02240-f002:**
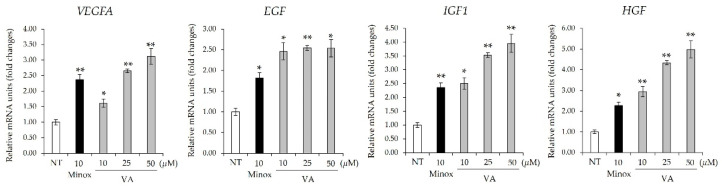
Veratric acid increases expression of growth factors. Expression levels of *VEGFA*, *EGF*, *IGF1*, and *HGF* are shown. All values were measured in triplicate (n = 3). Data are presented as mean ± SD. *, *p* < 0.05; **, *p* < 0.01.

**Figure 3 ijms-26-02240-f003:**
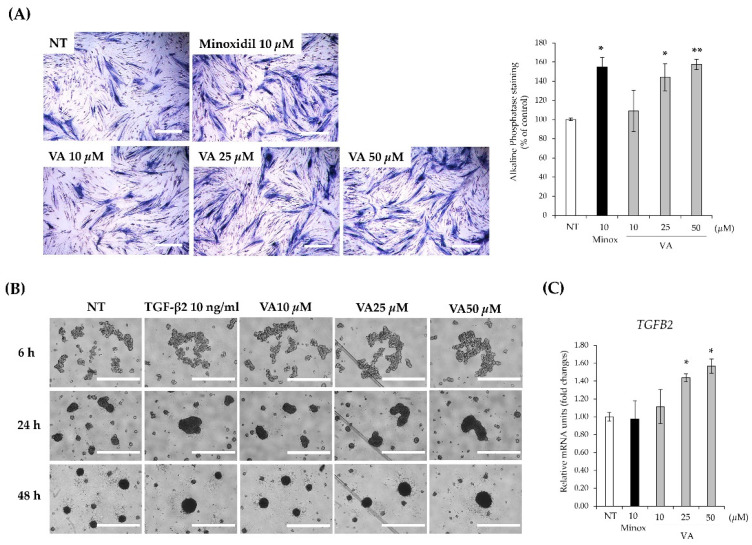
Veratric acid increases hair inductivity-related factors. (**A**) Alkaline phosphatase staining of HFDPCs cultured with VA or minoxidil and graph of values measured using ImageJ. Scale bar = 1 μm. (**B**) Degree of aggregation of HFDPCs. Scale bar = 1000 μm. (**C**) Expression level of *TGFB2* mRNA in HFDPCs. All values were measured in triplicate (n = 3). Data are presented as mean ± SD. *, *p* < 0.05; **, *p* < 0.01.

**Figure 4 ijms-26-02240-f004:**
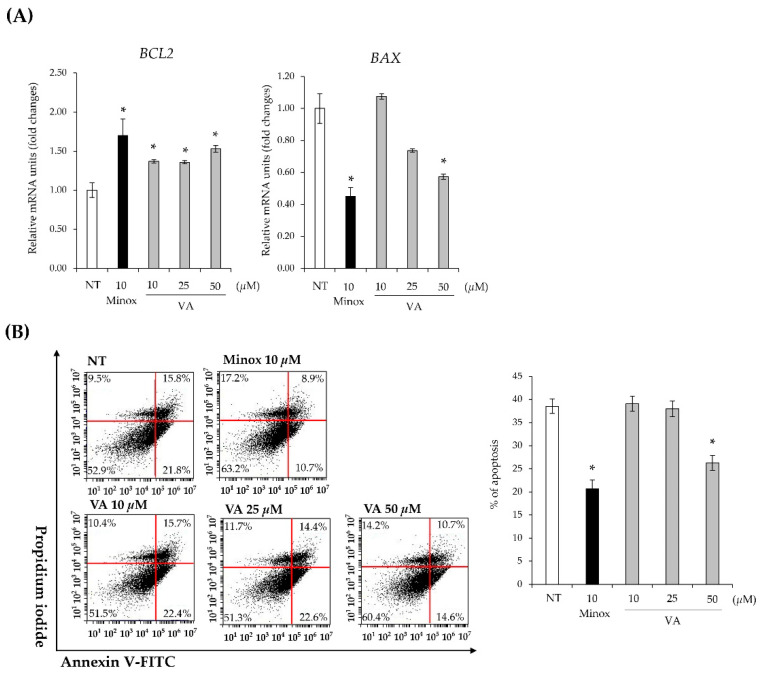
Veratric acid decreases apoptosis of HFDPCs. (**A**) Expression levels of *BCL2* and *BAX* mRNAs in HFDPCs cultured with VA or minoxidil. *, *p* < 0.05. (**B**) Flow cytometry analysis of apoptosis of HFDPCs by Annexin V FITC and PI staining. *, *p* < 0.001. All values were measured in triplicate (n = 3). Data are presented as mean ± SD.

**Figure 5 ijms-26-02240-f005:**
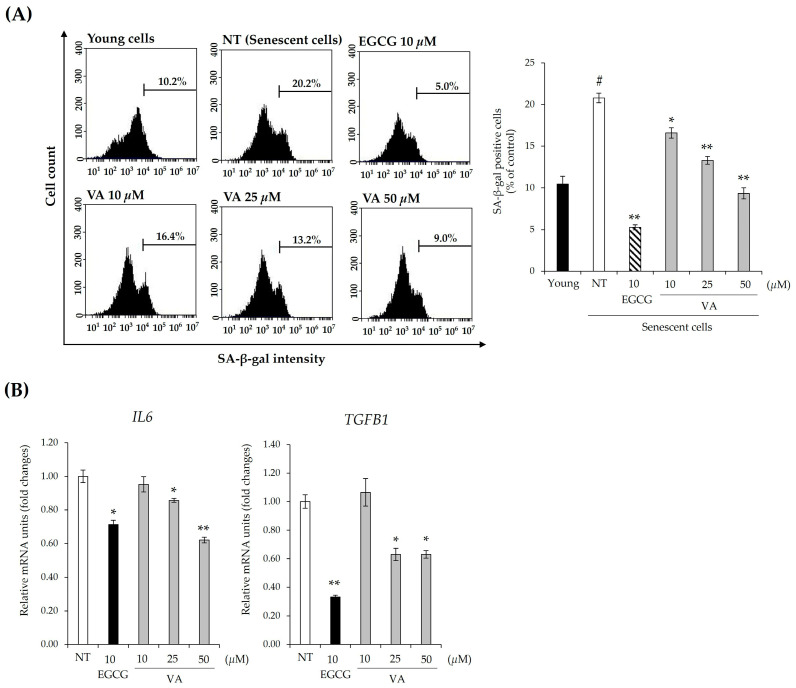
Veratric acid reduces senescence in replicative senescent HFDPCs. (**A**) Flow cytometry analysis of β-gal activity in replicative senescent HFDPCs cultured with VA or EGCG. #, *p*-value < 0.01 vs. the young group; *, *p*-value < 0.01; **, *p*-value < 0.01 vs. the replicative senescence-induced untreated group (control group). (**B**) Expression levels of *IL6* and *TGFB1* mRNAs in replicative senescent HFDPCs. *, *p*-value < 0.05; **, *p*-value < 0.01. All values were measured in triplicate (n = 3). Data are presented as mean ± SD.

**Figure 6 ijms-26-02240-f006:**
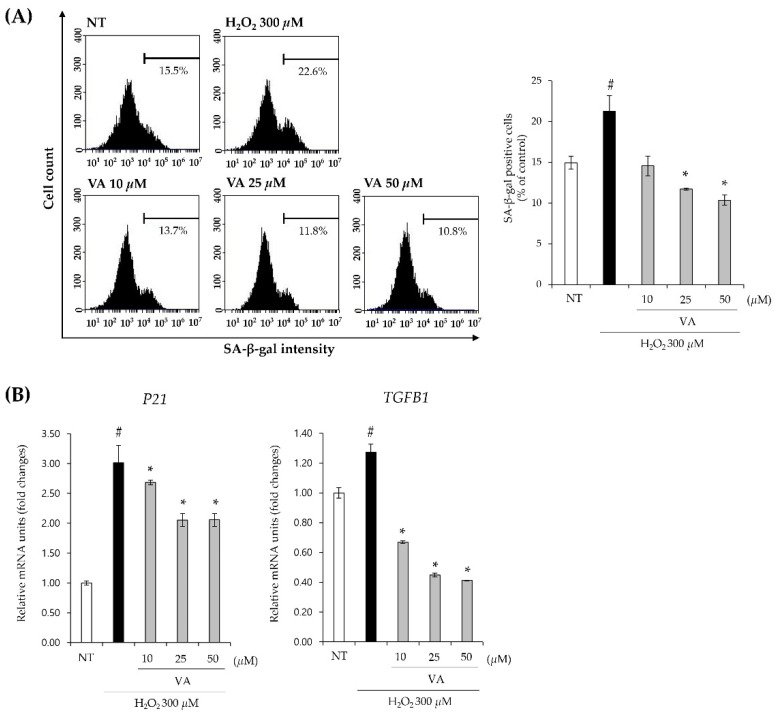
Veratric acid alleviates senescence induced by oxidative stress on HFDPCs. (**A**) Flow cytometry analysis of β-gal activity. After pretreatment with 300 μM H_2_O_2_ for 2 h, cells were incubated with VA for 72 h. β-gal activity was measured by FACS. (**B**) Expression levels of *P21* and *TGFB1* mRNAs in HFDPCs. All values were measured in triplicate (n = 3). Data are presented as mean ± SD. #, *p*-value < 0.05 vs. H_2_O_2_-untreated group; *, *p*-value < 0.05 vs. H_2_O_2_-treated group.

**Figure 7 ijms-26-02240-f007:**
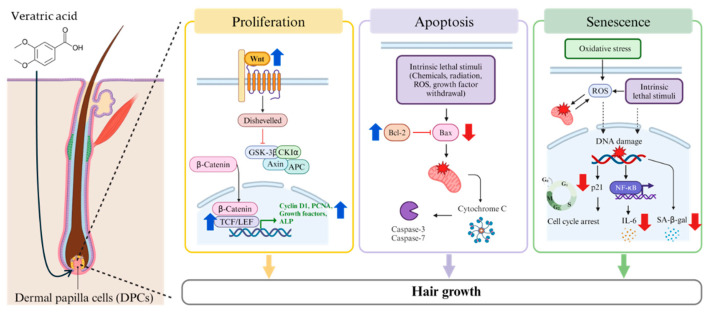
Hair growth mechanism of veratric acid in HFDPCs.

**Table 1 ijms-26-02240-t001:** Sequences of primers used for real-time PCR (5′ to 3′).

Gene	Forward Primer	Reverse Primer
*EGF*	TGCCAGCTGCACAAATACAGA	TCTTACGGAATAGTGGTGGTCATC
*HGF*	GAGAGTTGGGTTCTTACTGCACG	CTCATCTCCTCTTCCGTGGACA
*IGF1*	AGGAAGTACATTTGAAGAACGCAACT	CCTGCGGTGGCATGTCA
*VEGF* *A*	TTGCCTTGCTGCTCTACCTCCA	GATGGCAGTAGCTGCGCTGATA
*C* *CN* *D1*	TCTACACCGACAACTCCATCCG	TCTGGCATTTTGGAGAGGAAGTG
*L* *EF* *1*	CTACCCATCCTCACTGTCAGTC	GGATGTTCCTGTTTGACCTGAGG
*TGF* *B* *1*	TACAACCCGTGTTGCTCTC	GTTGCTGAGGTATCGCCAGGAA
*TGF* *B* *2*	AAGAAGCGTGCTTTGGATGCGG	ATGCTCCAGCACAGAAGTTGGC
*B* *CL* *2*	ATCGCCCTGTGGATGACTGAGT	GCCAGGAGAAATCAAACAGAGGC
*B* *AX*	TCAGGATGCGTCCACCAAGAAG	TGTGTCCACGGCGGCAATCATC
*IL6*	AGACAGCCACTCACCTCTTCAG	TTCTGCCAGTGCCTCTTTGCTG
*P21*	AGGTGGACCTGGAGACTCTCAG	TCCTCTTGGAGAAGATCAGCCG
*GAPDH*	GTCTCCTCTGACTTCAACAGCG	ACCACCCTGTTGCTGTAGCCAA

## Data Availability

All data included in this study are available upon request by contact with the corresponding author.

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
