# Peer review of "Anti-Hair Loss Effect of Veratric Acid on Dermal Papilla Cells"

_ijms, 2025, doi:10.3390/ijms26052240_

Round 1
Reviewer 1 Report
Comments and Suggestions for Authors
Journal: IJMS (ISSN 1422-0067)
Manuscript ID: ijms-3496158
Type: Article
Title: Anti-hair loss effect of Veratric acid on dermal papilla cells
The manuscript titled " Anti-hair loss effect of Veratric acid on dermal papilla cells” by Jiyoung You et al, includes a well-structured and comprehensive examination into the possible therapeutic benefits of Veratric Acid (VA) on hair loss. It describes the primary experimental methodologies utilized to evaluate the many processes underpinning VA's activity on Hair Follicle Dermal Papilla Cells (HFDPCs), including its effects on proliferation, hair inductivity, apoptosis, and cellular senescence.
Abstract:
1: Mention in the abstract if VA's effects were time- or dose-dependent.
2: Include a more direct link between VA's mechanisms (e.g., apoptosis suppression, anti-inflammatory effects) and its function in hair loss prevention rather than merely HF regeneration in the introduction section?
3: The introduction focuses on VA's antioxidant, anti-inflammatory, and lipid-lowering capabilities in general, but it does not explain how these actions assist dermal papilla cells.
Discuss how VA affects important pathways in DPCs related to hair development, including as Wnt/β-catenin signaling, BMP, and TGF-β.
Material and method:
Cell culture:
Inconsistent Cultural Media Across Sections
4: Indicate passage numbers used?
5: Uses follicular dermal papilla cell growth medium (Promocell).
6: Cell Viability Assay
Uses DMEM with 1% FBS.
What media was actually utilized in the experiments?
Low FBS (1%) levels may generate starvation-like effects, lowering viability findings.
Make it clear which medium was used for each experiment, and justify the use of 1% FBS in the viability assay.
7: Justify the 2h incubation time or increase it to 3–4 hours.
Explain how absorbance was normalized.
Provide a reference for Minoxidil as a positive control.
8: Quantitative Real-Time PCR.
There is no description of how the RNA purity and concentration were determined (A260/A280 ratio).
There are no details about PCR cycling conditions.
9: 2. Results 64
2.1. Veratric Acid Increases Proliferation of HFDPCs
How does Veratric Acid (VA) affect the Wnt/β-catenin signaling pathway in HFDPCs?
What function does LEF1 play in the VA-induced proliferation of HFDPCs?
Does VA directly activate β-catenin or modify upstream regulators of the Wnt pathway?
Discussion:
10: How does VA use particular signaling pathways to reduce β-gal activity and prevent senescence in replicatively senescent HFDPCs?
11: Does VA just affect HFDPC growth and survival through the Wnt/β-catenin pathway, or does it also affect other signaling cascades?
12: How does VA affect the migratory ability of HFDPCs under oxidative stress as compared to untreated cells?
13: What biological pathways relate VA's activation of cyclin D1 and LEF1 to cell cycle progression in HFDPCs?
Author Response
Thank you for your comments; they have helped us improve the quality of the manuscript. The comments are listed below with our responses. All changes were marked in red color.
Comments 1: Mention in the abstract if VA's effects were time- or dose-dependent.
Response 1: Thank you for your insightful comment. We have clarified that VA's effects are dose-dependent on cell viability, proliferation, ALP activity, aggregation, apoptosis inhibition, and senescence suppression in HFDPCs. The Abstract has been revised to explicitly state these dose-dependent effects. The revised text can be found on Page 1, Line 16-20.
Comments 2: Include a more direct link between VA's mechanisms (e.g., apoptosis suppression, anti-inflammatory effects) and its function in hair loss prevention rather than merely HF regeneration in the introduction section?
Response 2: Thank you for your insightful comment. We have revised the introduction to establish a more direct connection between VA’s mechanisms, particularly its anti-inflammatory and anti-apoptotic effects, and its potential role in hair loss prevention, rather than solely focusing on HF regeneration.
Previous studies suggest that VA exerts anti-inflammatory and anti-apoptotic effects that contribute to HF maintenance and hair loss prevention. VA has been reported to suppress CXCL1, a pro-inflammatory chemokine secreted from dermal white adipose tissue (dWAT), which is known to delay HF cycling, induce premature catagen transition, and promote HF miniaturization. Additionally, CXCL1 has been shown to induce apoptosis in the hair matrix and connective tissue sheath, further impairing follicle maintenance. By inhibiting CXCL1 expression, VA may help preserve HF integrity and support hair growth, suggesting its potential therapeutic role in preventing hair loss.
These revisions have been incorporated into the introduction, which now explicitly links VA’s biological effects to its role in hair loss prevention. The revised text can be found on Page 2, Lines 66–73.
Comments 3: The introduction focuses on VA's antioxidant, anti-inflammatory, and lipid-lowering capabilities in general, but it does not explain how these actions assist dermal papilla cells.
Discuss how VA affects important pathways in DPCs related to hair development, including as Wnt/β-catenin signaling, BMP, and TGF-β.
Response 3: Thank you for your valuable comment. Accordingly, we have revised the introduction to include additional descriptions of the effects of VA on key signaling pathways related to hair follicle growth.
Inflammatory cytokines, such as TNF-α and IL-1β, activate the NF-κB signaling pathway, which can suppress Wnt/β-catenin signaling, a key regulator of hair growth, and induce apoptosis in hair follicle stem cells. Additionally, oxidative stress inhibits Wnt/β-catenin signaling and promotes senescence in DPCs by activating p16 and p53, thereby disrupting the hair growth cycle. Therefore, the previously reported anti-inflammatory and antioxidant effects of VA are expected to suppress inflammatory mediators and reduce oxidative stress, thereby protecting Wnt/β-catenin signaling, inhibiting apoptosis and senescence, and ultimately promoting DPC activation and hair growth.
Additionally, VA’s lipid-lowering effects, which is less directly relevant to hair follicle physiology, have been excluded. The revised content can be found on Page 2, Line 54-65.
Material and method:
Comments 4: Indicate passage numbers used?
Response 4: We have clarified the passage numbers used in our experiments. All experiments, except for the replicative senescence model, were conducted using HFDPCs at passages 3 or 4. This information has been added to the Cell Culture section to ensure consistency.
The revised text can be found on Page 10, Lines 334.
Comments 5: Uses follicular dermal papilla cell growth medium (Promocell).
Response 5: We have clarified the culture conditions in the Cell Culture section. HFDPC maintenance was performed using follicle dermal papilla cell growth medium (PromoCell), while all experiments involving VA treatment were conducted in DMEM supplemented with 1% serum and penicillin/streptomycin. This ensures consistency across the study and has been explicitly stated in the revised manuscript.
The revised text can be found on Page 10, Lines 332-334.
Comments 6: Cell Viability Assay
Uses DMEM with 1% FBS.
What media was actually utilized in the experiments?
Low FBS (1%) levels may generate starvation-like effects, lowering viability findings. Make it clear which medium was used for each experiment, and justify the use of 1% FBS in the viability assay.
Response 6: The use of 1% FBS in the viability assay was intentional to minimize external growth stimulation and allow for a controlled evaluation of VA's effects on cell viability. While low FBS levels can induce mild nutrient deprivation, this condition is widely employed in similar studies to distinguish compound-induced effects from serum-driven proliferation​ [Ref.1]. The referenced study highlights how low-serum conditions influence cell viability and signaling pathways, emphasizing the role of serum-reduced media in minimizing experimental interference and improving reproducibility. This demonstrates that reducing FBS concentration helps minimize external influences and enables a clearer assessment of the test compound’s effects.
As mentioned in our response to Comment 5, details regarding the media used for each experiment have been included in the Cell Culture section to ensure consistency.
The revised text can be found on Page 10, Lines 332–334.
[Ref.1] Rashid, M.U.; Coombs, K.M. Serum-reduced media impacts on cell viability and protein expression in human lung epithelial cells. J. Cell. Physiol. 2019, 234, 7718-7724.
Comments 7: Justify the 2h incubation time or increase it to 3–4 hours.
Response 7-1: We previously confirmed that MTT assay results at 2 hours and 3–4 hours showed no significant differences. Based on these findings, we set the incubation time to a maximum of 2 hours to optimize experimental efficiency while ensuring reliable measurements.
Explain how absorbance was normalized
Response 7-2: The mean absorbance of the control (NT) group was set to 100%, and all measured absorbance values were normalized to this standard and expressed as a percentage. This information has been added to Page 11, Line 355-357 in the manuscript.
Provide a reference for Minoxidil as a positive control.
Response 7-3: Thank you for your valuable comment. We have provided a reference that supports the use of Minoxidil as a positive control for DPC proliferation. While the concentration used in our study may differ slightly from those in previous reports, we acknowledge that the effective concentration of Minoxidil can vary depending on experimental conditions. Based on our preliminary dose-response experiments, 10 µM Minoxidil demonstrated the most significant effect under our specific experimental conditions and was therefore used in this study. The added content can be found on Page 11, Line 348-350 in the manuscript.
Comments 8: Quantitative Real-Time PCR.
There is no description of how the RNA purity and concentration were determined (A260/A280 ratio).
Response 8-1: Total RNA was measured using the Gen5 microplate reader software ver 2.09 (Bio-Tek Inc.) with the nucleic acid quantification function. The RNA purity was determined by the A260/A280 ratio, with an average value of 1.98, and the RNA concentration averaged 800 ng/µL. For cDNA synthesis, an average of 1 µg of RNA was used. This information has been added to Page 11, Line 373-376 in the manuscript.
There are no details about PCR cycling conditions.
Response 8-2: We have added detailed information on the qRT-PCR cycling conditions in the Methods section of the manuscript. The qRT-PCR protocol consisted of an initial denaturation at 95°C for 10 minutes, followed by 40 cycles of denaturation at 95°C for 15 seconds and annealing/extension at 60°C for 35 seconds. After amplification, a dissociation curve analysis was performed with the following conditions: 95°C for 15 seconds, 60°C for 1 minute, and 95°C for 15 seconds.
This information has been added to Page 11, Line 382-389 in the manuscript.
Comments 9: (2. Results)
2.1. Veratric Acid Increases Proliferation of HFDPCs
How does Veratric Acid (VA) affect the Wnt/β-catenin signaling pathway in HFDPCs?
Response 9-1: Thank you for your insightful question. Our results indicate that VA increases β-catenin protein levels and upregulates Wnt target genes, such as LEF1 and Cyclin D1, in HFDPCs. These findings suggest that VA may activate the Wnt/β-catenin signaling pathway, contributing to enhanced HFDPC proliferation. However, the exact molecular mechanism by which VA regulates β-catenin remains unclear. The revised text can be found on Page 2, Lines 88-99 and Page 8, Lines 227–234.
What function does LEF1 play in the VA-induced proliferation of HFDPCs?.
Response 9-2: LEF1 is a key transcription factor in the Wnt/β-catenin signaling pathway. Its increased expression in response to VA treatment suggests that VA promotes Wnt-mediated transcriptional activity. Since Cyclin D1 is a downstream target of Wnt signaling and plays a crucial role in cell cycle progression, our findings support the idea that VA-induced LEF1 upregulation facilitates HFDPC proliferation by driving the expression of cell cycle regulators. The revised text can be found on Page 2, Lines 88-99 and Page 8, Lines 227–234,
Does VA directly activate β-catenin or modify upstream regulators of the Wnt pathway?
Response 9-3: While our results show that VA increases β-catenin protein levels, it remains unclear whether VA directly activates Wnt signaling by modulating upstream regulators such as GSK-3β or stabilizes β-catenin through alternative mechanisms. Further studies are needed to determine the precise molecular mechanisms underlying VA-induced β-catenin accumulation and its potential regulation of upstream Wnt pathway components.
We have further clarified this aspect in the Results and Discussion sections. Additionally, we have included a statement highlighting the need for further research to determine whether VA influences upstream regulators of the Wnt/β-catenin signaling pathway or promotes β-catenin nuclear translocation to regulate target gene expression. The revised text can be found on Page 9, Lines 237-240.
Discussion:
Comments 10: How does VA use particular signaling pathways to reduce β-gal activity and prevent senescence in replicatively senescent HFDPCs?
Response 10: Thank you for your insightful comment. Our findings showed that VA significantly reduced β-gal activity and suppressed IL-6 and TGF-β1 expression in replicatively senescent HFDPCs. Given that IL-6 and TGF-β1 are key components of the senescence-associated secretory phenotype (SASP), these results suggest that VA may modulate SASP-related signaling pathways to attenuate senescence.
IL-6 expression is primarily regulated by the NF-κB signaling pathway, which is persistently activated in senescent cells ​. Chronic NF-κB activation contributes to the secretion of proinflammatory cytokines, including IL-6, which exacerbates cellular senescence. Thus, VA may exert its anti-senescence effects by suppressing NF-κB activity, leading to reduced IL-6 expression.
TGF-β1 is a major upstream regulator of SASP and cellular senescence, often activating the Smad2/3 pathway to induce cell cycle arrest​. Smad2/3 activation has been shown to upregulate p16 and p21 expression, thereby promoting senescence. VA-mediated suppression of TGF-β1 suggests that VA may interfere with Smad signaling, thereby preventing the reinforcement of senescence.
Although our study did not directly assess these pathways, further investigations are warranted to determine whether VA modulates NF-κB or TGF-β/Smad signaling to regulate SASP expression and senescence progression in HFDPCs.
The revised text can be found on Page 10, Lines 288–297.
Comments 11: Does VA just affect HFDPC growth and survival through the Wnt/β-catenin pathway, or does it also affect other signaling cascades?
Response 11: Thank you for your insightful comment. Our results indicate that VA influences HFDPC function beyond Wnt/β-catenin signaling.
VA enhanced hair inductivity by increasing ALP activity, cell aggregation, and TGF-β2 expression, while also upregulating key growth factors (VEGF, EGF, IGF1, HGF). Additionally, VA promoted cell survival by increasing BCL2 and decreasing BAX, suggesting anti-apoptotic effects.
Furthermore, VA reduced β-gal activity and SASP factors (IL-6, TGF-β1), indicating potential modulation of NF-κB or TGF-β/Smad pathways to suppress senescence.
These findings suggest that VA regulates HFDPC proliferation, survival, and hair inductivity through multiple pathways beyond Wnt/β-catenin. Further studies are needed to clarify these mechanisms.
Comments 12: How does VA affect the migratory ability of HFDPCs under oxidative stress as compared to untreated cells?
Response 12: Thank you for your insightful comment. We did not evaluate the effect of VA on the migratory ability of HFDPCs under oxidative stress in this study.
However, HFDPC migration plays a critical role in hair follicle regeneration, and previous studies have shown that oxidative stress can impair cell migration by disrupting cytoskeletal dynamics [Ref. 1]. Given that VA enhanced HFDPC survival and upregulated Wnt/β-catenin signaling, which is involved in both proliferation and migration, VA may also influence cell mobility under oxidative stress [Ref. 2].
Further investigations, such as wound healing assays or transwell migration assays, would be necessary to clarify the effects of VA on HFDPC migration. This would provide deeper insights into the role of VA in hair follicle regeneration under oxidative stress conditions.
[Ref. 1] Wilson, C.;, González-Billault, C. Regulation of cytoskeletal dynamics by redox signaling and oxidative stress: implications for neuronal development and trafficking. Front. Cell. Neurosci. 2015, 9, 381.
[Ref. 2] Liu, J.; Xiao, Q.; Xiao, J.; Niu, C.; Li, Y.; Zhang, X.; Zhou, Z.; Shu, G.; Yin, G. Wnt/β-catenin signalling: function, biological mechanisms, and therapeutic opportunities. Signal Transduct. Target. Ther. 2022, 7, 3.
Comments 13: What biological pathways relate VA's activation of cyclin D1 and LEF1 to cell cycle progression in HFDPCs?
Response 13: Thank you for your insightful comment. Our results indicate that VA promotes HFDPC proliferation by upregulating Cyclin D1 and LEF1, both of which are key regulators of cell cycle progression.
However, this study did not directly evaluate the effect of VA on cell cycle progression. Previous studies have shown that LEF1 functions as a transcription factor in the Wnt/β-catenin signaling pathway, directly regulating Cyclin D1 expression, while Cyclin D1 promotes G1/S transition by activating CDK4/6, leading to Rb phosphorylation and E2F-mediated transcription of S-phase genes.
Given that VA upregulated both LEF1 and Cyclin D1 in HFDPCs, it is possible that VA influences cell cycle progression through Wnt/β-catenin signaling. However, further studies, such as flow cytometry-based cell cycle analysis, are required to confirm this effect.
To further investigate this aspect, we will consider additional experiments to determine whether VA directly affects cell cycle progression in HFDPCs.
The revised text can be found on Page 2, Lines 88–94.

Reviewer 2 Report
Comments and Suggestions for Authors
Minor comments:
Line 127-132: How does Veratric Acid (VA) influence apoptosis in HFDPCs, and what role do BCL2 and BAX play in this process?
Line 79-81: What is the role of Veratric Acid in modulating the Wnt/β-catenin signaling pathway, and how does this contribute to hair growth?
Line 91-96: How does VA affect the expression of key hair growth factors such as VEGFA, EGF, IGF1, and HGF?
Line 105-115: What role does Veratric Acid play in hair follicle self-aggregation and the restoration of hair-inductive activity?
Line 166-178: Does VA have any impact on oxidative stress-induced premature senescence of HFDPCs, and how is this mechanism regulated?
Line 187-188: How does Veratric Acid (VA) influence the transition between hair cycle phases (anagen, catagen, telogen)?
Line 244-247: What experimental models were used to validate the efficacy of VA on hair loss, and how reliable are these models?
Line 70-72: How does VA compare to minoxidil in terms of its effects on hair growth and HFDPC activation?
Author Response
Thank you for your comments; they have helped us improve the quality of the manuscript. The comments are listed below with our responses. All changes were marked in red color.
Comments 1: Line 127-132: How does Veratric Acid (VA) influence apoptosis in HFDPCs, and what role do BCL2 and BAX play in this process??
Response 1: Thank you for your insightful comment. Our results indicate that VA inhibits apoptosis in HFDPCs by modulating the expression of key apoptotic regulators, BCL2 and BAX. BCL2 is an anti-apoptotic protein that prevents mitochondrial outer membrane permeabilization (MOMP), thereby inhibiting cytochrome c release and caspase activation. In contrast, BAX is a pro-apoptotic protein that promotes MOMP, leading to apoptotic cell death.
In our study, VA treatment upregulated BCL2 while downregulating BAX expression, suggesting that VA shifts the balance of apoptotic signaling toward cell survival (Figure 4A). Consistently, FACS analysis demonstrated a significant reduction in apoptotic cell populations in the VA-treated group compared to the control (Figure 4B). These findings indicate that VA protects HFDPCs from apoptosis, potentially enhancing their viability and function in hair follicle regeneration.
This mechanistic insight has been incorporated into the Results section on Page 5, Line 153-158, 161-162.
Comments 2: Line 79-81: What is the role of Veratric Acid in modulating the Wnt/β-catenin signaling pathway, and how does this contribute to hair growth?
Response 2: Thank you for your insightful comment. Our results suggest that VA promotes HFDPC proliferation by modulating the Wnt/β-catenin signaling pathway. Specifically, VA treatment increased the expression levels of β-catenin, LEF1, and Cyclin D1, key components of the Wnt/β-catenin pathway (Figure 1C, 1D). LEF1 is a transcription factor that, when activated by β-catenin, enhances the transcription of genes involved in cell cycle progression, including Cyclin D1, which promotes the G1-to-S phase transition. The increased expression of PCNA, a marker of DNA replication, further supports the proliferative effects of VA on HFDPCs.
The activation of Wnt/β-catenin signaling in DPCs plays a crucial role in hair follicle regeneration by maintaining DPC identity, promoting dermal-epidermal interactions, and inducing anagen phase entry. Given that VA enhances the expression of Wnt target genes, it may support hair growth by reinforcing DPC proliferation and function, which are essential for hair follicle regeneration.
However, it remains unclear whether VA directly activates Wnt signaling by increasing ligand expression or by inhibiting negative regulators such as GSK-3β, thereby stabilizing β-catenin. Further studies are necessary to elucidate the precise mechanism by which VA regulates Wnt/β-catenin signaling. This revision has been incorporated into the Results section on Page 2, Line 88-99 and the Discussion section on Page 8, Line 229-237.
Comments 3: Line 91-96: How does VA affect the expression of key hair growth factors such as VEGFA, EGF, IGF1, and HGF?
Response 3: Thank you for your valuable comment. In this study, we confirmed that VA upregulated the expression of VEGFA, EGF, IGF1, and HGF in HFDPCs in a dose-dependent manner.
The Wnt/β-catenin signaling pathway is known to play a crucial role in regulating dermal papilla cell function and the expression of various growth factors. It is possible that VA-induced Wnt/β-catenin activation contributed to the observed increase in these growth factors. Therefore, VA may promote hair growth by modulating the Wnt/β-catenin signaling pathway to enhance growth factor expression.
The relevant modifications have been incorporated into the Results sections on Page 3, Line 113-117 and Discussion sections on Page 9, Line 249-259.
Comments 4: Line 105-115: What role does Veratric Acid play in hair follicle self-aggregation and the restoration of hair-inductive activity?
Response 4: Thank you for your insightful comment. In this study, we found that VA enhanced HFDPC self-aggregation and increased the expression of TGF-β2 in a concentration-dependent manner.
TGF-β2 has been identified as a key regulator of DPC self-aggregation, which is closely associated with follicle neogenesis and hair growth induction. Additionally, self-aggregated DPCs exhibit enhanced hair-inductive properties, resembling embryonic dermal condensates that contribute to follicle development.
Given that TGF-β2 plays a crucial role in promoting DPC self-aggregation, the observed increase in TGF-β2 expression following VA treatment suggests that VA facilitates self-aggregation through TGF-β2 upregulation. This enhanced self-aggregation may, in turn, contribute to the restoration of hair-inductive activity in DPCs, supporting follicle regeneration.
The relevant modifications have been incorporated into the Results sections on Page 4, Line 127-140 and Discussion sections on Page 9, Line 266-268, 269-275.
Comments 5: Line 166-178: Does VA have any impact on oxidative stress-induced premature senescence of HFDPCs, and how is this mechanism regulated?
Response 5: Thank you for your insightful comment. Our study demonstrated that VA significantly inhibited β-gal activity in Hâ‚‚Oâ‚‚-induced senescent HFDPCs, indicating its protective effect against oxidative stress-induced premature senescence. Additionally, VA downregulated the expression levels of p21 and TGF-β1, both of which were upregulated under oxidative stress conditions.
p21 is a critical cell cycle inhibitor that mediates senescence by halting the G1/S and G2/M transitions through p53 activation. The suppression of p21 by VA suggests that VA mitigates oxidative stress-induced senescence by preventing p53-dependent cell cycle arrest.
Additionally, TGF-β1 is known to maintain the senescent phenotype by reinforcing SASP expression. Given that VA significantly reduced TGF-β1 expression, it is possible that VA alleviates senescence by disrupting the TGF-β1-mediated signaling loop, thereby inhibiting the reinforcement of senescence-associated pathways.
To avoid redundancy, we have reflected this interpretation in the Discussion section (Page 10, Line 303-319).
Comments 6: Line 187-188: How does Veratric Acid (VA) influence the transition between hair cycle phases (anagen, catagen, telogen)?
Response 6: Thank you for your insightful comment. In this study, we demonstrated that VA promotes HFDPC proliferation and upregulates key growth factors such as VEGFA, IGF1, and HGF. Since HFDPCs play a critical role in maintaining hair follicle homeostasis and initiating the anagen phase, these findings suggest that VA may contribute to the hair cycle transition by supporting DPC function.
Additionally, Wnt/β-catenin signaling, which was upregulated by VA in our study, is known to regulate hair cycle progression by maintaining the anagen phase and facilitating the transition from telogen to anagen. Given that VA enhanced Wnt/β-catenin activity and increased the expression of anagen-associated growth factors, it is possible that VA promotes hair cycle progression through these mechanisms.
However, our study did not directly assess the effects of VA on specific hair cycle phases. Further investigations, including in vivo hair cycle analysis, would be necessary to clarify whether VA actively modulates the transition between anagen, catagen, and telogen phases.
We have incorporated this discussion into Page 8, Line 234-237 in the revised manuscript.
Comments 7: Line 244-247: What experimental models were used to validate the efficacy of VA on hair loss, and how reliable are these models?
Response 7: Thank you for your insightful question. In this study, HFDPCs were used as the primary in vitro model to evaluate the effects of VA on hair follicle-associated cells. HFDPCs are widely recognized as a reliable model for studying hair follicle biology, as they play a crucial role in hair growth regulation, dermal-epidermal interactions, and extracellular signaling within the hair follicle niche.
While our study demonstrated that VA enhances HFDPC proliferation, aggregation, and growth factor expression, further studies utilizing more complex models, such as 3D spheroid culture systems, ex vivo hair follicle organ culture, or in vivo models, would strengthen the translational relevance of our findings.
The use of HFDPCs alone provides valuable mechanistic insights into VA's effects at the cellular level, but we acknowledge the need for future investigations employing multicellular and in vivo models to further validate the efficacy of VA in hair follicle regeneration and hair loss prevention.
This information has been incorporated into the Discussion section on Page 10, Line 320-325.
Comments 8: Line 70-72: How does VA compare to minoxidil in terms of its effects on hair growth and HFDPC activation?
Response 8: Thank you for your insightful question. In this study, minoxidil was used as a positive control to compare its effects with VA on HFDPC function. Both VA and minoxidil promoted cell proliferation, as demonstrated by MTT and Ki67 staining assays, showing comparable results in this aspect.
However, VA exhibited distinct effects compared to minoxidil in pathways related to hair inductivity. Specifically, VA significantly increased the expression of TGF-β2, a key factor associated with hair follicle induction. In contrast, minoxidil did not induce a similar increase in TGF-β2 expression.
Therefore, while both VA and minoxidil enhance HFDPC proliferation, VA may more effectively support hair follicle inductivity by upregulating TGF-β2 expression. These findings suggest that VA and minoxidil may act through different mechanisms in hair follicle regeneration.
